# gga-miR-449b-5p Regulates Steroid Hormone Synthesis in Laying Hen Ovarian Granulosa Cells by Targeting the IGF2BP3 Gene

**DOI:** 10.3390/ani12192710

**Published:** 2022-10-09

**Authors:** Xing Wu, Na Zhang, Jing Li, Zihao Zhang, Yulong Guo, Donghua Li, Yanhua Zhang, Yujie Gong, Ruirui Jiang, Hong Li, Guoxi Li, Xiaojun Liu, Xiangtao Kang, Yadong Tian

**Affiliations:** 1College of Animal Science and Technology, Henan Agricultural University, Zhengzhou 450046, China; 2Henan Key Laboratory for Innovation and Utilization of Chicken Germplasm Resources, Zhengzhou 450046, China

**Keywords:** laying hens, gga-miR-449b-5p, granulosa cell, proliferation, steroid synthesis, IGF2BP3

## Abstract

**Simple Summary:**

The aim of this study was to explore the regulatory effect of gga-miR-449b-5p on GC proliferation and steroidogenesis in laying hens. Our results showed that gga-miR-449b-5p had no effect on the proliferation of GCs, but regulated the expression of key genes involved in steroid synthesis and the secretion of P4 and E2. In addition, gga-miR-449b-5p could target IGF2BP3 and inhibit its mRNA and protein expression. Therefore, we concluded that gga-miR-449b-5p played an important role in the synthesis of steroid hormones in laying hens.

**Abstract:**

MiRNAs have been found to be involved in the regulation of ovarian function as important post-transcriptional regulators, including regulators of follicular development, steroidogenesis, cell atresia, and even the development of ovarian cancer. In this study, we evaluated the regulatory role of gga-miR-449b-5p in follicular growth and steroid synthesis in ovarian granulosa cells (GCs) of laying hens through qRT-PCR, ELISAs, western blotting and dual-luciferase reporter assays, which have been described in our previous study. We demonstrated that gga-miR-449b-5p was widely expressed in granulosa and theca layers of the different-sized follicles, especially in the granulosa layer. The gga-miR-449b-5p had no significant effect on the proliferation of GCs, but could significantly regulate the expression of key steroidogenesis-related genes (StAR and CYP19A1) (*p* < 0.01) and the secretion of P4 and E2 (*p* < 0.01 and *p* < 0.05). Further research showed that gga-miR-449b-5p could target IGF2BP3 and downregulate the mRNA and protein expression of IGF2BP3 (*p* < 0.05). Therefore, this study suggests that gga-miR-449b-5p is a potent regulator of the synthesis of steroid hormones in GCs by targeting the expression of IGF2BP3 and may contribute to a better understanding of the role of functional miRNAs in laying hen ovarian development.

## 1. Introduction

The egg production performance of laying hens depends on the function of ovaries and the developmental ability of follicles, which are regulated by a complex and delicate network. Many studies have shown that the synthesis of steroid hormones in follicular theca cells (TCs) and granulosa cells (GCs) is essential for the regulation of follicular development and maturation, cell proliferation, differentiation and apoptosis [1,2,3,4].

The hypothalamic-pituitary-ovarian axis plays an important role in the reproduction of animals by regulating steroid hormone synthesis. However, numerous studies have proven that ovarian steroidogenesis is also regulated by various factors in the ovary, including key steroidogenesis-related genes, hormones and other regulatory factors. In general, hormones can exert their biological effects only when they specifically combine with their receptors to form hormone-receptor complexes. Progesterone (PR), androgen (AR) and estrogen (ER) receptors are located in the nucleus of GCs and TCs and bind specifically to hormones to form hormone receptor complexes, which are involved in follicular growth and development, maturation and ovulation, as well as the synthesis of steroids [5,6,7,8]. The genes such as steroidogenic acute regulatory protein (StAR), cytochrome P450 family 11 subfamily A member 1 (CYP11A1), 3β-hydroxysteroid dehydrogenase (3β-HSD), and cytochrome P450 family 19 subfamily A member 1 (CYP19A1) are directly involved in the synthesis of progesterone and estrogen [9,10,11,12]. In addition, several regulatory factors, such as IGF-1, EGF, TGF-β, and SF-1, and transcription factors (CATA-4/6, FOXL2, WT-1, DAX-1, AP-1, and SP-1) [13,14,15] regulate the expression of ovarian steroid synthesis genes as well as hormone secretion by activating multiple molecular signaling pathways. Thus, it is necessary to explore the further regulatory mechanism of steroid synthesis for follicular development and ovulation in chickens.

In recent years, many studies have reported that noncoding RNAs, such as miRNAs, lncRNAs and circRNAs, play major roles in the regulation of the reproductive function of laying hens [16,17,18,19,20]. MiRNA is a kind of noncoding small 22 nt RNA that mostly regulates gene expression at the post-transcriptional level. By binding to the miRNA response element (MRE) in the 3′UTR, which is the target of mRNA, miRNAs act as negative modulators of gene expression, inhibit or silence target gene expression, and regulate mRNA and protein expression at the mRNA and protein levels [21,22]. MiRNAs are widely distributed in various tissues and cell types and participate in a variety of biological regulatory processes, playing an important regulatory role in the occurrence and development of diseases, tumors and cancers [23,24,25]. At present, GCs have been the main target cells for the study of ovarian function related to miRNA regulation, particularly in mammals. Many studies have shown that functional miRNAs play an important role in regulating ovarian function, follicular development and atresia, cell proliferation and apoptosis, steroid hormone synthesis and even ovarian cancer [26,27,28,29]. In mammals, miR-383, miR-323-3p, miR-320a, miR-130a-3p, miR-1246, miR-31 and miR-20b influence steroid hormone synthesis in granulosa cells [30,31,32,33,34,35], and miR-214-3p, miR-324-3p and miR-335-5p promote granulosa cell proliferation in the ovary [36,37,38]. Unfortunately, in poultry, the study of functional miRNAs in ovarian development and function of laying hens still lags behind. Only a few studies have proven that miR-26a-5p, miR-1b-3p and miR-23b-3p play an important role in the regulation of follicular development and steroid synthesis in chickens [39,40,41].

In our previous study, we predicted that gga-miR-449b-5p might be involved in the regulation of proliferation and steroid synthesis in ovaries of hens via transcriptome sequencing analysis of hen ovarian tissue during the four classic physiological periods (15 w, 20 w, 30 w, and 68 w), which represent initial ovarian development, sexual maturation, the peak laying period and the late laying period [42]. In another previous study, we predicted that IGF2BP3 may be its targeted regulatory gene [20]. At present, the reports on this gene are mainly focused on cell development, proliferation and migration [43,44], especially in cancer development and progression [45,46,47], but there are few reports on the reproduction of laying hens. Therefore, in the present study, we explored the regulatory effect of gga-miR-449b-5p on cell proliferation and steroid synthesis and hormone secretion in GCs and explored the relationship between gga-miR-449b-5p and IGF2BP3 in GCs to further explore the role of gga-miR-449b-5p in ovarian function in laying hens.

## 2. Materials and Methods

### 2.1. Ethics Approval

All studies involving Hy-Line brown laying hens were approved by the regulators for the administration of affairs concerning experimental animals (Revised Edition, 2017). The protocols have been reviewed and approved by the Henan Agricultural University Institutional Animal Care and Use Committee (Permit Number: 19−0068).

### 2.2. Sample Collection

Forty healthy Hy-Line brown laying hens were collected at the age of 30 weeks and euthanized by cervical dislocation, and then the whole ovaries of six laying hens were removed. The follicles were divided into the following ten groups: small white follicles with a diameter < 4 mm (SWF), large white follicles with a diameter of 4−6 mm (LWF), small yellow follicles with a diameter of 6–8 mm (SYF), large yellow follicles with a diameter of 9–12 mm (LYF), and preovulatory follicles with a diameter > 12 mm (F6-F1) [48]. The obtained groups of follicles were placed in PBS buffer containing 3% double antibiotics to remove the residual connective tissue and attached blood filaments. The outer membrane layer was peeled off with curved forceps, the follicles were cut in half with scissors and the end was gently shaken with forceps until all the white granulosa layer came out; the remaining layer is the theca layer. After the granulosa layer or theca layer were mixed and divided equally into 6 biological replicates, they were immediately used for gene expression analysis. The small yellow follicles with a diameter of 6−8 mm were removed from the remaining 34 hens and divided into 4 groups: NC group, gga-miR-449b-5p mimic group, NCR group and gga-miR-449b-5p inhibitor group, followed by cell proliferation assay, ELISA assay and western blotting assay.

### 2.3. Cell Culture

The granulosa layer of small yellow follicles with a diameter of 6–8 mm was collected from the remaining 34 hens according to the method described above, and washed three times in PBS buffer. All GCs were assembled in 1.5 mL centrifuge tubes, ground into a homogeneous paste by a cell grinding rod, mixed in 15 mL centrifuge tubes and digested in a 37 °C environment with 0.25% trypsin of the same volume for 10 min. After the digestion process was completed, single cells were obtained by 200 µm filtration. The cells were assembled by 1800 rpm centrifugation at room temperature for 5 min. After repeated centrifugation twice, the suspended cells in complete culture medium containing 2.5% fetal bovine serum and 1% double antibiotics were spread in a 12-well plate, which was incubated in a cell incubator at 37 °C and 5% CO_2_ for 12 h and then the cells were transfected, as previously reported [49].

### 2.4. RNA Extraction, cDNA Synthesis, and Quantitative Real-Time PCR (qRT-PCR)

Total RNA was extracted from GCs and TCs using TRIzol reagent (Novizan, Nanjing, China). The purity and concentration of the RNA was determined by measuring the ratio between the absorbance at 260 nm and 280 nm by a spectrophotometer (Thermo, Waltham, MA, USA). All samples were of acceptable purity (the range of the absorbance ratio from 1.9 to 2.1). cDNA was synthesized by reverse transcription with the HiScrip^®®^III First Strand cDNA Synthesis Kit (Novizan, Nanjing, China) containing gDNA wiper. MiRNA was reverse transcribed by a HiScrip^®®^III RT SuperMix for qPCR kit (Novizan, Nanjing, China). The samples were stored frozen at −20 °C and a ChamQ Universal SYBR qPCR Master Mix (Vazyme, Nanjing, China) was used for real-time quantitative polymerase chain reaction (qRT-PCR). The β-actin gene was used as the reference gene for mRNA, and U6 was used as the reference gene for miRNA. The relative quantification of related genes and miRNAs was performed by the 2^−∆∆Ct^ method [50]. The primer sequence information is listed in Table 1.

### 2.5. Plasmid Construction

The mimics and inhibitors of gga-miR-449b-5p and their negative controls (NC and NCR) were synthesized by RiboBio (Guangzhou, China). The MMP2 3′UTR fragment with binding sites was cloned into the psiCHECK-2 dual luciferase reporter vector by PCR amplification. To construct a mutant MMP2 3′UTR vector, we designed mutant primers, and the sequences are shown in Table 1.

### 2.6. Cell Transfection and Treatment

Lipofectamine 3000 reagent (Invitrogen, Carlsbad, CA, USA) was used to transfect miRNA mimics, and a ribo FECT CP transfection kit (Guangzhou, China) was used to transfect miRNA inhibitors. The transfection experiment was carried out according to the concentration recommended by the manufacturer when the density of follicular granulosa cells in a 12-well plate was ≥ 70%. The new complete medium was changed after transfection for 4 h.

### 2.7. Cell Proliferation Assay

The GCs were spread in 12-well plates for the EdU assay. GC proliferation was measured using a Cell Proliferation EdU Image Kit (Abbkine, Wuhan, China) after transfection for 24 h. Nuclear staining experiments were performed with 4′,6-diamidino-2-phenylindole (DAPI, Invitrogen, Carlsbad, California USA) after two hours of incubation in the incubator. Finally, images were captured with a fluorescence microscope (Olympus Ts2-FL, Olympus, Shinjuku-ku, Tokyo, Japan).

GCs were evenly spread in a 96-well plate, placed in a 37 °C incubator containing 5% CO_2_ and cultured until they were transfected. The 96-well plates were removed at 12 h, 24 h, 36 h and 48 h. According to the manufacturer’s instructions, 100 µL of the original medium was removed, while 100 µL of complete medium containing 10% CCK-8 reagent (Dojindo, Kumamoto, Japan) was added to the incubator for an additional 2 h. Afterward, samples were collected for enzyme labeling (BioTek, Winooski, VT, USA), and absorbance was detected (OD value) at a wavelength of 450 nm.

### 2.8. Flow Cytometric Analysis

GCs were collected 24 h after transfection and washed twice with PBS. DNA was incubated with PI (Solarbio, Beijing, China) staining solution at 4 °C for 30 min. GCs were analyzed by adjusting the excitation wavelength of the flow cytometer (BD Biosciences, San Jose, CA, USA) to Ex = 488 nm and the emission wavelength to Em = 530 nm.

### 2.9. ELISA for Steroid Hormones

GCs were transfected in 12-well plates for 24 h, and cell supernatants were collected. Concentrations of progesterone (P4), testosterone (T), and estradiol (E2) were determined by the Chicken P4, T, and E2 ELISA Kit (Jiangsu Meimian Industrial Co., Ltd., Jiangsu, China), respectively, according to the manufacturer’s instructions (The sensitivity of ELISA Kits of P4, T, E2 were typically less than 10 pmol/L, 1.0 pg/mL and 0.1 pg/mL; tolerance within batch and tolerance between batches of CV < 10% and no cross-reactivity for all three ELISA kits.).

### 2.10. Western Blotting Assay

The proteins from GCs were extracted at 36 h post-transfection with a RIPA lysis buffer (Beyotime, Shanghai, China). Primary antibodies for IGF2BP3 and GAPDH were purchased from Novusbio (Littleton, CO, USA) and Affinity (Cincinnati, OH, USA), respectively, and were incubated with samples overnight at 4 °C. The secondary antibody, HRP-conjugated goat anti-rabbit antibodies provided by Elabscience (Wuhan, China), were incubated at room temperature for 1 h. Finally, the optical density values of the target band were analyzed with the Odyssey FC NIR Protein Processing System (LI-COR, Lincoln, NE, USA).

### 2.11. Dual-Luciferase Reporter Assay

The DF-1 cell line is the most investigated and widely used chicken cell line, and its culture is the same as that reported by Himly et al.; these cells were used for the dual-luciferase reporter assay [51]. When the DF-1 cell density was ≥ 70%, the gga-miR-449b-5p mimics or NC was cotransfected with IGF2BP3 WT or IGF2BP3 MUT and MMP2 WT or MMP2 MUT for 36 h using Lipofectamine 3000 reagent (Invitrogen, Carlsbad, CA, USA). Firefly and sea kidney luciferase activities were detected by the Dual Luciferase Reporter Gene Assay System Kit (Promega, Madison, WI, USA) according to the manufacturer’s instructions.

### 2.12. Statistical Analysis

All data were statistically analyzed by SPSS package version 22.0 and are presented as the means ± SEMs. A *p* value < 0.05 indicated a statistically significant difference. GraphPad Prism 7.0 software (GraphPad Software, Inc., San Diego, CA, USA) was used for visualization of all data for statistical purposes.

## 3. Results

### 3.1. Differential Expression of gga-miR-449b-5p in TCs and GCs at All Levels

We investigated the distribution of gga-miR-449b-5p in the TCs and GCs of follicles of different sizes by qRT−PCR. The results showed that the expression level of gga-miR-449b-5p was significantly higher in follicular GCs than in TCs, especially follicles with a diameter of 4–6 mm, 6–8 mm, 9–12 mm and preovulatory follicles with a diameter > 12 mm (F6-F3) (*p* < 0.01 and *p* < 0.05). In addition, the highest expression was found in prehierarchical GCs (Figure 1).

### 3.2. gga-miR-449b-5p Has No Effect on the Proliferation of GCs

We investigated the role of gga-miR-449b-5p in the proliferation of GCs using qRT-PCR, CCK-8 assays, EdU assays and flow cytometry. The results showed that the transfection efficiency of gga-miR-449b-5p was increased approximately 600-fold in GCs by the gga-miR-449b-5p mimic (*p* < 0.01) and significantly diminished by the gga-miR-449b-5p inhibitor (*p* < 0.01; Figure 2a). Then, a qRT-PCR assay was used to detect the mRNA expression of the proliferation-related genes CCND1, CCND2, CDK1, CDK2 and CDK6. We found that the gga-miR-449b-5p mimic suppressed the mRNA expression of CCND1 and CCND2 (*p* < 0.05 and *p* < 0.05) but upregulated the mRNA expression of CDK2 and CDK6 (*p* < 0.01; Figure 2b). Next, a CCK-8 analysis was performed to determine the changes in GC viability at 12, 24, 36 and 48 h, and we found that both overexpression of and interference with gga-miR-449b-5p had no significant effect on the viability of GCs (Figure 2c,d).

EdU analysis was performed to detect the number of proliferating GCs after transfection of cells with gga-miR-449b-5p mimics and gga-miR-449b-5p inhibitors. We found that neither overexpression of nor interference with gga-miR-449b-5p had a significant effect on the proliferation of GCs (Figure 2e). The same results were found for the flow cytometric assays (Figure 2f,g). These results indicated that gga-miR-449b-5p had no effect on the viability and proliferation of GCs in laying hen follicles.

### 3.3. gga-miR-449b-5p Regulates Steroid Secretion by GCs

We determined the role of gga-miR-449b-5p in P4, T, and E2 secretion in chicken granulosa cells using qPCR and ELISAs. The expression of key genes related to steroid synthesis was first detected. The qPCR results showed that the overexpression of gga-miR-449b-5p reduced the mRNA expression of StAR and CYP19A1 (*p* < 0.01), and interfering with gga-miR-449b-5p had the opposite effect (Figure 3a). The ELISA results showed that the P4 and E2 levels in the GCs after transfection with gga-miR-449b-5p mimic were decreased (*p* < 0.01 and *p* < 0.05), while the P4 and E2 levels in the GCs after transfection with gga-miR-449b-5p inhibitor were increased (*p* < 0.05 and *p* < 0.01; Figure 3b). These results suggest that gga-miR-449b-5p can regulate the synthesis and secretion of steroids and then affect the ovarian function of laying hens.

### 3.4. IGF2BP3 Is a gga-miR-449b-5p Target

We examined several potential targets predicted in previous studies to be associated with the regulatory effects of gga-miR-449b-5p on steroidogenesis, including IGFBP4, PGRMC1, MMP2, IGF2BP3, BMP3 and E2F5. We found that the mRNA expression of MMP2 and IGF2BP3 decreased markedly after overexpression of gga-miR-449b-5p (*p* < 0.05 and *p* < 0.01; Figure 4a). To verify which was the direct target gene of gga-miR-449b-5p, we used a dual-luciferase reporter assay. The results showed that there was no targeting relationship between MMP2 and gga-miR-449-5p (Figure 4b). Notably, the gga-miR-449b-5p mimic significantly decreased the activity of WT IGF2BP3 (*p* < 0.01), but no significant changes were noted for the MUT, which indicated that IGF2BP3 was directly targeted by gga-miR-449b-5p (Figure 4c).

### 3.5. Expression of IGF2BP3 Is Regulated by gga-miR-449b-5p

To test the validity of the putative target, we transfected gga-miR-449b-5p mimic or NC and inhibitor or NCR into chicken GCs. The results showed that compared with the gga-miR-449b-5p inhibitor NCR, the gga-miR-449b-5p inhibitor significantly increased the protein expression of IGF2BP3 (*p* < 0.05), while compared with the gga-miR-449b-5p mimic NC, the gga-miR-449b-5p mimic significantly inhibited the protein expression of IGF2BP3 (*p* < 0.01). These results further suggested that gga-miR-449b-5p plays a regulatory role by targeting IGF2BP3 (Figure 5).

## 4. Discussion

The development of ovarian follicles is the basis of female animal reproduction. As the most basic functional unit of the ovary, the granulosa layer of poultry follicles is closely linked to the development and selection of dominant follicles. The proliferation, apoptosis and steroid synthesis of follicular granulosa cells are not only affected by nutritional and environmental factors but also regulated by miRNAs [52].

The miR-449b-5p has been a known regulatory function in human reproduction-related diseases, such as breast cancer, endometrial cancer and cervical cancer. Jiang et al. found that miR-449b-5p may inhibit the growth and invasion of breast cancer cells by inhibiting the CREPT/Wnt/β-catenin axis [53]. Zhao et al. reported that miR-449b-5p inhibited the proliferation of endometrial cancer cells by targeting MDM4 [54]. Another report found that overexpression of miR-449b-5p in cervical cancer cell lines significantly inhibits cell proliferation [55]. All the above results show that miR-449b-5p has a potentially important influence on regulation of cell proliferation, but its role in ovarian granulosa cells needs to be further explored. Therefore, we speculate that gga-miR-449b-5p has a similar role in GCs. CCND1, CCND2, CDK1, CDK2 and CDK6 were reported to be marker genes for cell proliferation, which was found in GCs [56,57]. In this study, we examined the role of gga-miR-449b-5p in GC proliferation by marker gene-related proliferation, CCK-8, flow cytometry and EdU experiments. Surprisingly, we found that the proliferation of GCs was not affected by gga-miR-449b-5p mimic or inhibitor transfection, indicating that the function of gga-miR-449b-5p in GCs is different from that of the above studies, and this difference can be attributed to the differences between species and cell models. As gga-miR-449b-5p was confirmed to be enriched in the pathway related to steroid synthesis, we explored the specific role of gga-miR-449b-5p in steroid synthesis in GCs.

P4, androgen and E2 are the main steroids that play a crucial role in the regulation of female fertility [58]. Previous reports identified an androgen receptor in GCs [59]. GCs are believed to be mediated by follicle stimulating hormone and synthesize P4 under the action of StAR, CYP11A1 and 3β-HSD [60,61]. In addition, P4 is a precursor of estradiol synthesis in TCs, which is stimulated by luteinizing hormone and catalyzed by CYP19A1. Thus, the synthesis of steroids in TCs needs to be carried out with the participation of GCs to regulate the synthesis and secretion of steroids [62,63,64,65,66]. Therefore, in this experiment, we determined P4, T and E2 secretion by ELISAs as well as the expression of StAR, CYP11A1, 3β-HSD, and CYP19A1 by qPCR. The results showed that gga-miR-449b-5p inhibited the expression of the steroid synthesis-related genes StAR, 3β-HSD and CYP19A1 and the production of P4 and E2. This result is consistent with the previous results concerning the validation of miRNA function in steroid the synthesis of GCs [35,37,43].

Several studies have shown that the insulin-like growth factor-2 mRNA-binding protein family is involved in mammalian follicular development and steroid secretion [67,68,69]. IGF2BP3 is an important member of this family [70]. Current studies have shown that IGF2BP3 ensures the early embryonic development of zebrafish by maintaining the stability of maternal RNA [71]. The expression of IGF2BP3 in medaka is also closely associated with oocyte development [72]. These results indicate that IGF2BP3 may play a role in the development of animal ovaries. We further investigated several potential target genes of gga-miR-449b-5p in this study. Interestingly, it was found that overexpression of gga-miR-449b-5p could significantly reduce the expression of MMP2 and IGF2BP3. Furthermore, through double luciferase reporter gene detection, we demonstrated that gga-miR-449b-5p was able to target IGF2BP3. Furthermore, IGF2BP3 gene and protein levels decreased after gga-miR-449b-5p mimic transfection, and vice versa. These data suggest that gga-miR-449b-5p regulates steroid synthesis in GCs by targeting IGF2BP3.

## 5. Conclusions

gga-miR-449b-5p inhibits the secretion of P4 and E2 in GCs by targeting the IGF2BP3 gene and inhibiting its expression. Our results may provide scientific insights into the regulatory mechanism of miRNAs in follicular development in the future.

## Figures and Tables

**Figure 1 animals-12-02710-f001:**
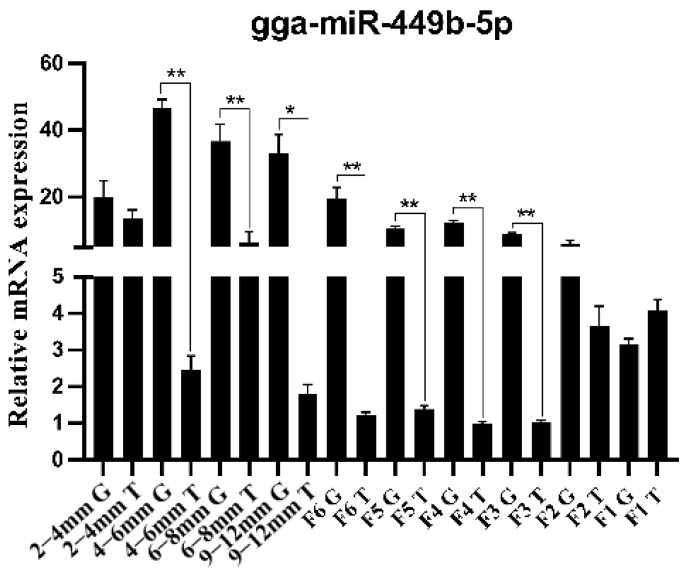
Expression of gga-miR-449b-5p in the TCs and GCs of follicles of different sizes. ** and * indicated that there were significant differences (*p* < 0.01) and significant differences (*p* < 0.05), respectively.

**Figure 2 animals-12-02710-f002:**
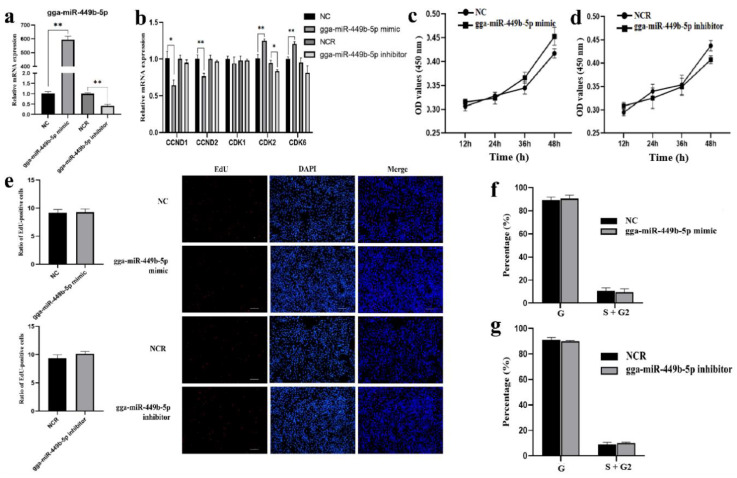
gga-miR-449b-5p has no effect on the proliferation of GCs in small yellow follicles (6-8 mm). (**a**) The transfection efficiency of gga-miR-449b-5p mimics or inhibitor; (**b**) The mRNA expression of key genes related to proliferation of GCs following transfection of gga-miR-449b-5p mimics or inhibitor; (**c**,**d**) Cell growth curves determined by the CCK-8 assay following transfection with gga-miR-449b-5p mimics or inhibits in granulosa cells; (**e**) Proliferation of GCs was assayed using EdU assays after transfection with gga-miR-449b-5p mimics or inhibitor; (**f**,**g**) Cell cycle changes of GCs after transfection with gga-miR-449b-5p mimics or inhibitor as shown by flow cytometry. ** and * indicated that there were significant differences (*p* < 0.01) and significant differences (*p* < 0.05), respectively.

**Figure 3 animals-12-02710-f003:**
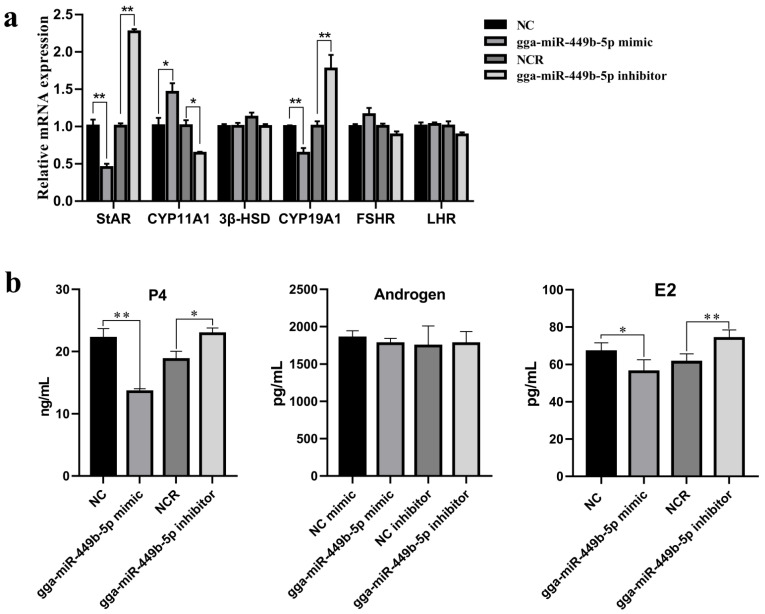
gga-miR-449b-5p regulates steroid secretion from GCs of small yellow follicles (6–8 mm). (**a**) The mRNA expression of key genes related to steroid synthesis in GCs after transfection with gga-miR-449b-5p mimics or inhibitor. (**b**) The concentrations of steroid hormones in GCs after transfection with gga-miR-449b-5p mimics or inhibitor. ** and * indicated that there were significant differences (*p* < 0.01) and significant differences (*p* < 0.05), respectively.

**Figure 4 animals-12-02710-f004:**
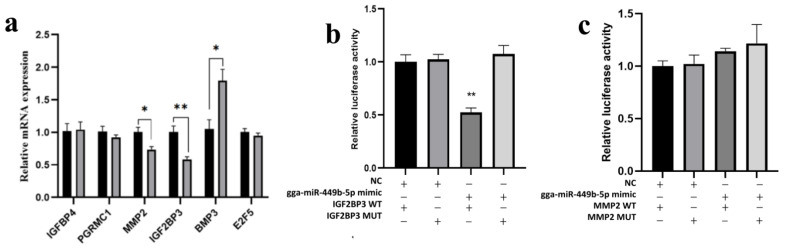
IGF2BP3 is directly targeted by gga-miR-449b-5p. (**a**) The mRNA expression of potential target genes following transfection with gga-miR-449b-5p mimics; (**b**) Dual-luciferase reporter assay of the DF-1 cell line cotransfected with gga-miR-449b-5p mimics and IGF2BP3 3′ UTR-WT or IGF2BP3 3′ UTR-MUT; (**c**) Dual-luciferase reporter assay of the DF-1 cell line cotransfected with gga-miR-449b-5p mimics and MMP2 3′ UTR-WT or MMP2 3′ UTR-MUT. ** and * indicated that there were significant differences (*p* < 0.01) and significant differences (*p* < 0.05), respectively.

**Figure 5 animals-12-02710-f005:**
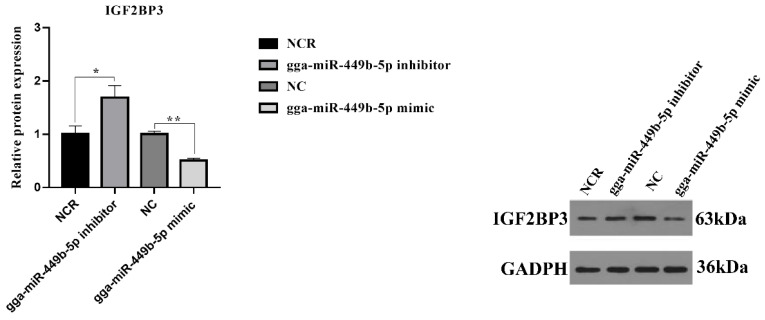
The protein expression of IGF2BP3 in GCs of small yellow follicles (6−8 mm) transfected with gga-miR-449b-5p mimic or inhibitor. ** and * indicated that there were significant differences (*p* < 0.01) and significant differences (*p* < 0.05), respectively.

**Table 1 animals-12-02710-t001:** Primer sequences.

Primer Name	Primer Sequence (5′–3′)	Product Length (bp)	Genbank Number
*CCND1*	F: ATAGTCGCCACTTGGATGCT	122	NM_205381
	R: AACCGGCTTTTCTTGAGGGG		
*CCND2*	F: TCCGGAAACATGCACAAACG	257	XM_015292118.2
	R: CCGGACTTGCCTAAGGTTGC		
*CDK1*	F: TGGCCTTGAACCACCCATAC	147	NM_205314.1
	R: AGGCAGGCAGGCAAAGATAA		
*CDK2*	F: ACGTGATCCACACGGAGAAC	132	NM_001199857
	R: GCAGCTGGAACAGGTAGCTC		
*CDK6*	F: AGCAGCCCAGAAGAGATGATT	132	NM_001007892.2
	R: GAGAAATACGCACAAACCCTGT		
*StAR*	F: GTCCCTCGCAGACCAAGT	196	NM_204686
	R: TCCCTACTGTTAGCCCTGA		
*CYP11A1*	F: GTGGACACGACTTCCATGACT	174	NM_001001756
	R: GAGAGTCTCCTTGATGGCGG		
*3β-HSD*	F: TGGAAGAAGATGAGGCGCTG	185	NM_205118
	R: GGAAGCTGTGTGGATGACGA		
*CYP19A1*	F: GGCCTCCAGCAGGTTGAAAG	214	NM_001001761.3
	R: ATAGGCACTGTGGCAACTGG		
*FSHR*	F: GAGCGAGGTCTACATACA	281	NM_205079
	R: GCACAAGCCATAGTCA		
*LHR*	F: GGGCTTTCCCAAGCCTACAT	133	NM_204936.2
	R: TGGTGTCTTTATTGGCGGCT		
*IGFBP4*	F: AACTTCCACCCCAAGCAGT	123	NM_204353.1
	R: GCAATCCAAGTCCCCCTTCA		
*PGRMC1*	F: AGATCGTGGGCTCACCTCTA	157	NM_001271939.1
	R: AGCTGCTCCAGTGTGAAGTC		
*MMP2*	F: CGATGCTGTCTACGAGTCCC	96	NM_204420.2
	R: TAGCCCCTATCCAGGTTGCT		
*IGF2BP3*	F: TCCTGGTGAAGACGGGCTAC	133	XM_015281444.4
	R: CTTTTAGGGACCGAATGCTC		
*BMP3*	F: ACAGGGCAAAGAGTAAGAAAAAG	136	NM_001034819.2
	R: AGATAGCGTCGGGCACAATA		
*E2F5*	F: GCCTTCCAGACTCAGTGTTG	148	NM_001030942.1
	R: GGCTCCTCCATCTTTGCTAT		
*β* *-actin*	F: CAGCCAGCCATGGATGATGA	147	NM_205518.2
	R: ACCAACCATCACACCCTGAT		
*IGF2BP3 WT*	F: ccgctcgagTTACATAACACTGCCATGAATA	244	-
	R: ataagaatgcggccgcAGTCCGTAGTACTCCTGGCTGG		
*IGF2BP3 MUT*	F: ccgctcgagTTACATAATGACATAGTGAATAACCTAAGGGA	244	-
	R: ataagaatgcggccgcAGTCCGTAGTACTCCTGGCTGG		
*MMP2 WT*	F: ccgctcgagCGAGTTTGATCATTACTGCCA	337	-
	R: ataagaatgcggccgcGAAAGCCTAACCAAACAAAAC		
*MMP2 MUT*	F: ccgctcgagCGAGTTTGATCATTGACATTGTTTATTTACATAAT	337	-
	R: ataagaatgcggccgcGAAAGCCTAACCAAACAAAAC		

## Data Availability

No publicly archived dataset for this experiment.

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
