# Peer review of "gga-miR-449b-5p Regulates Steroid Hormone Synthesis in Laying Hen Ovarian Granulosa Cells by Targeting the IGF2BP3 Gene"

_animals, 2022, doi:10.3390/ani12192710_

Round 1
Reviewer 1 Report
Reviewer’s Comments (Animals)
This study ‘‘gga-miR-449b-5p regulates steroid hormone synthesis in laying hen ovarian granulosa cells by targeting the IGF2BP3 gene ’’ was appropriately designed and the conclusion was supported by the experimental data. In general, the article is interesting and well presented. However, before final acceptance, the following issues need to be addressed.
1. Abstract
- Please, include the p-values.
- Include the significance of the study to the conclusion section.
2. Materials and methods
- Sample Collection
- The Materials and Methods lack experimental design section. Please, add.
- Line 119 - 120: ‘’The relative quantification of related genes and miRNAs was performed by the 2-∆∆Ct method’’. Please, cite a reference.
- Table 1: The primer name in the primer sequences should be Italicized.
- Cell culture
- Please, describe how the granular layer was removed ?
- Please, change CO2 to CO2, Change throughout the manuscript.
- RNA extraction, cDNA synthesis, and quantitative real-time PCR (qRT‒PCR)
- Was the concentration of the RNA determined? Please, add it in the Materials and Method section.
- Please, include the primer product lengths and accessory numbers.
- Unify the units used throughout the manuscript.
- ELISA for steroid hormones
- Please, include the sensibility range of the kits used.
- Results
- Differential expression of gga-miR-449b-5p in TCs and GCs at all levels
- Please, include p-values.
- Line 186 - 189. Please, rephrase.
- The graphs (Figure 1) does not show any statistical difference indicators. Please, modify accordingly.
- 3.2 Line 192: Rephrase the sub-title.
- gga-miR-449b-5p has no effect on GCs
- Flow cytometry protocol was not given in Materials and Methods section. Please, add.
- The figure legend of Figure 2 should be put under the respective Figure.
- The abbreviations should come immediately after the conclusion section.
General Comments
1. Minor English check needed, especially with sentence construction and punctuation.
2. All sub-titles should be re-checked and re-written.
3. Please, improve on the quality of all the figures throughout the manuscript (especially, Figure 2 and Figure 4).
Author Response
Response to Reviewer 1 Comments
Abstract
Point 1: Please, include the p-values.
Response 1: Thank you very much for your advice. We have looked through the whole abstract section and changed “…key steroidogenesis-related genes (StAR and CYP19A1) and significantly suppress the secretion of P4 and E2. Further research showed that gga-miR-449b-5p could target IGF2BP3 and downregulate the mRNA and protein expression of IGF2BP3.” to “… key steroidogenesis-related genes (StAR and CYP19A1) (P<0.01) and significantly suppress the secretion of P4 and E2 (P<0.01 and P<0.05). Further research showed that gga-miR-449b-5p could target IGF2BP3 and downregulate the mRNA and protein expression of IGF2BP3 (P<0.05).”
Point 2: Include the significance of the study to the conclusion section.
Response 2: Thank you very much for your advice. I have added the significance of the study in the conclusion as follows “and may contribute to a better understanding of the role of functional miRNAs in laying hen ovarian development.”
Materials and methods
Point 3: Sample Collection The Materials and Methods lack experimental design section. Please, add.
Response 3: Thank you very much for your advice. I have added it as follows “The small yellow follicles with a diameter of 6-8 mm were removed from the remaining 34 hens and divided into 4 groups: NC group, gga-miR-449b-5p mimic group, NCR group and gga-miR-449b-5p inhibitor group, followed by cell proliferation assay, ELISA assay and western blotting assay.”
Point 4: Line 119 - 120: ‘’The relative quantification of related genes and miRNAs was performed by the 2-∆∆Ct method’’. Please, cite a reference.
Response 4: Thank you very much for your advice. References have been provided. “[45] Livak KJ, Schmittgen TD. Analysis of relative gene expression data using real-time quantitative PCR and the 2− ΔΔCT method. methods. 2001;25:402-8.”
Point 5: Table 1: The primer name in the primer sequences should be Italicized.
Response 5: Thank you very much for your advice. The primer names in Table 1 have been italicized.
Point 6: Cell culture Please, describe how the granular layer was removed ?
Response 6: Thank you very much for your advice. I have added the method of separating the granulosa layer in the sample collection section. “The obtained groups of follicles were placed in PBS buffer containing 3% double antibiotics to remove the residual connective tissue and attached blood filaments. The outer membrane layer was peeled off with curved forceps, the follicles were cut in half with scissors and the end was gently shaken with forceps until all the white granulosa layer came out...”
Point 7: Cell culture Please, change CO2 to CO2, Change throughout the manuscript.
Response 7: Thank you very much for your advice. “…and 5% CO2 for12 hours, and…” changed into “…and 5% CO2 for12 hours, and…”; 2.7. Cell proliferation assay “…incubator containing 5% CO2 and cultured…” changed into “…incubator containing 5% CO2 and cultured…”
Point 8: RNA extraction, cDNA synthesis, and quantitative real-time PCR (qRT‒PCR) Was the concentration of the RNA determined? Please, add it in the Materials and Method section.
Response 8: Thank you very much for your advice. Yes, the concentration of the RNA was determined. I have added “The purity and concentration of the RNA was determined by measuring the ratio between the absorbance at 260 nm and 280 nm by a spectrophotometer (Thermo, Waltham, MA, USA). All samples were of acceptable purity (The rang of the absorbance ratio from 1.9 to 2.1)” in the Materials and Method section
Point 9: RNA extraction, cDNA synthesis, and quantitative real-time PCR (qRT‒PCR) Please, include the primer product lengths and accessory numbers.
Response 9: Thank you for your advice, I have added the relevant information to Table 1.
Point 10: Unify the units used throughout the manuscript.
Response 10: Thank you very much for your advice. “…were stored frozen at -20 degrees Celsius ChamQ Universal SYBR qPCR…” changed into “…were stored frozen at -20℃ ChamQ Universal SYBR qPCR…”
Point 11: ELISA for steroid hormones Please, include the sensibility range of the kits used.
Response 11: Thank you very much for your advice. I have added the following information: “The sensitivity of ELISA Kits of P4, T, E2 were typically less than 10 pmol/L, 1.0 pg/mL and 0.1 pg/mL.”
Results
Point 12: Differential expression of gga-miR-449b-5p in TCs and GCs at all levels Please, include p-values.
Response 12: Thank you very much for your advice. I have added the p-values in the appropriate places
Point 13: Differential expression of gga-miR-449b-5p in TCs and GCs at all levels Line 186 - 189. Please, rephrase.
Response 13: Thank you very much for your advice. I have rephrased lines 186-189 as follows: “Expression of gga-miR-449b-5p in the TCs and GCs of follicles of different sizes. ** and * indicated that there were significant differences (P < 0.01) and significant differences (P < 0.05), respectively.”
Point 14: Differential expression of gga-miR-449b-5p in TCs and GCs at all levels The graphs (Figure 1) does not show any statistical difference indicators. Please, modify accordingly
Response 14: Thank you very much for your advice. I have marked the statistical difference indicators at the graph (Fig. 1).
Point 15: 3.2 Line 192: Rephrase the sub-title
Response 15: Thank you very much for your advice. “gga-miR-449b-5p has no effect on GCs” changed into “gga-miR-449b-5p has no effect on the proliferation of GCs”
Point 16: Flow cytometry protocol was not given in Materials and Methods section. Please, add
Response 16: Thank you very much for your advice. “Flow cytometric analysis GCs were collected 24 hours after transfection and washed twice with PBS. DNA was incubated with PI (Solarbio, China) staining solution at 4℃ for 30 minutes. GCs were analyzed by adjusting the excitation wavelength of the flow cytometer (BD Biosciences, California, USA) to Ex=488 nm and the emission wavelength to Em=530 nm.”
Point 17: The figure legend of Figure 2 should be put under the respective Figure.
Response 17: Thank you very much for your advice. The figure legend of Figure 2 has been placed under the respective Figure.
Point 18: The abbreviations should come immediately after the conclusion section.
Response 18: Thank you very much for your advice. The abbreviation has been followed by the conclusion section
General Comments
1. Minor English check needed, especially with sentence construction and punctuation.
Response 1: Thank you very much for your comments. We have asked our native English-speaking colleague corrected our manuscript, and hope the quality of the current version reaches required standard of your Journal.
2. All sub-titles should be re-checked and re-written.
Response 2: Thank you very much for your comments. We have re-checked the sub-title and re-changed it accordingly.
3. Please, improve on the quality of all the figures throughout the manuscript (especially, Figure 2 and Figure 4).
Response 3: Thank you very much for your comments. We have improved on the quality of all the figures throughout the manuscript (especially, Figure 2 and Figure 4).

Reviewer 2 Report
General opinion
This is interesting paper comprising several data concerning role of gga-miR-449b-5p in steroid hormone synthesis in chicken ovarian follicles. The results revealed that the investigated miR does not affect granulosa cell proliferation in the ovarian follicles but it significantly decreases mRNA expression of steroidogenic genes resulting from suppression of progesterone and estradiol secretion from these follicles. This work may be published in Animals following major revision.
Detailed remarks:
Abstract
Line 19: the abbreviated GCs should be explained, i.e. “granulosa cells (GCs)”
Line 22: “granular” should be changed into “granulosa”. Moreover, in the entire paper exchange “granular” to “granulosa”, and “granular layer” to “granulosa layer”
Lines 28-29: this sentence comprising a final conclusion should be changed as follows: “…that gga-miR-449b-5p is a potent regulator of the synthesis…”
Introduction
Line 36: the abbreviated TCs and GCs should be explained, i.e. “theca cells (TCs) and granulosa cells (GCs)”
Line 39: “…by regulating steroid hormones” change into “...regulating steroid hormone synthesis”
Line 40: “…steroid hormones are also...” change into “..ovarian steroidogenesis is also…”
Line 42: taking into account the course of steroidogenesis the order of these genes should be as follows: “StAR, CYP11A1, 3β-HSD, and CYP19A1”. This order should be continued of kept in the whole text of the paper. Moreover, in the line 42 the authors needs to explain the names of these genes or proteins. For instance, 3β-hydroxysteroid dehydrogenase (3β-HSD), etc.
Line 45: It should be as follows: Progesterone (PR), androgen (AR) and estrogen (ER) receptors are located…”
Lines 58-59: It should be as follows: miRNAs act as negative modulators of gene expression.”
Line 68: change “mir-31” to “miR-31”
Materials and methods
Lines 90-98: Here is described the process of isolation of ten different groups of follicles from 6 hen ovaries. Please describe isolation of granulosa layer and theca layer from these follicles since in the Fig. 1 there are presented results of gga-miR-449b-5p expression in the granulosa and theca layers of different groups of ovarian follicles.
Lines 122-123: Table 1. Change order of the steroidogenic genes as follows: StAR, CYP11A1, 3β-HSD, and CYP19A1.
Lines 153-156: the order of steroid hormones should be as follows: progesterone (P4), testosterone (T), and estradiol (E2) accordingly to course of the steroidogenesis process. Please exchange “androgen” into “testosterone”. Moreover, more data concerning sensitivity of the methods, antibody cross-reactivities, and inter- and intraassay coefficients of variations need to be included.
Results
In the Fig. 1a. are presented data on expression of gga-miR-449b-5p in different weeks of age, however, in the Materials and methods there is no information that samples were collected from chickens in different age. Therefore, this Fig.1a should be removed.
Fig. 3a. Change order of the genes in this Fig as follows: StAR, CYP11A1, 3β-HSD, CYP19A1, FSHR, LHR.
Fig. 3b. Change order of these figures as follows: P4, T, E2.
Fig. 2, 3, 4 and 5. Provide statistical information concerning meaning of “*” and “**”.
Fig. 3. Chart description: gga-miR-449b-5p regulates steroid secretion from GCs of small yellow follicles (6-8 mm). The same remark concerns the other figures.
Discussion
Lines 297-308: Provide the appropriate order of steroid hormones and steroidogenic genes.
Lines 303-304: It should be as follows: “…we determined P4, T and E2 secretion by ELISAs as well as the expression of …by qPCR.”
Lines 307-308: It should be as follows: “…with the previous results concerning the validation of…”
Lines 309-321: In this paragraph the authors discuss role of IGFBP3 in ovarian steroidogenesis and the effect of gga-miR-449b-5p on this gene expression. Information concerning role of this gene and protein should be included in the Introduction, especially in the aim of the study.
Abbreviations: It does not comprise all abbreviations.
Author Response
Response to Reviewer 2 Comments
Abstract
Point 1: Line 19: the abbreviated GCs should be explained, i.e. “granulosa cells (GCs)”
Response 1: Thank you very much for your advice. We have explained that GCs are granulosa cells (GCs).
Point 2: Line 22: “granular” should be changed into “granulosa”. Moreover, in the entire paper exchange “granular” to “granulosa”, and “granular layer” to “granulosa layer”
Response 2: Thank you very much for your advice. We have looked through the whole text and changed “granular” to “granulosa”, and “granular layer” to “granulosa layer”.
Point 3: Lines 28-29: this sentence comprising a final conclusion should be changed as follows: “…that gga-miR-449b-5p is a potent regulator of the synthesis…”
Response 3: Thank you very much for your advice. We have changed to “…that gga-miR-449b-5p is a potent regulator of the synthesis…”
Introduction
Point 4: Line 36: the abbreviated TCs and GCs should be explained, i.e. “theca cells (TCs) and granulosa cells (GCs)”
Response 4: Thank you very much for your advice. We have explained that TCs are theca cells (TCs) and GCs are granulosa cells (GCs).
Point 5: Line 39: “…by regulating steroid hormones” change into “...regulating steroid hormone synthesis”
Response 5: Thank you very much for your advice. We have changed “…by regulating steroid hormones” to “...regulating steroid hormone synthesis”.
Point 6: Line 40: “…steroid hormones are also...” change into “..ovarian steroidogenesis is also…”
Response 6: Thank you very much for your advice. We have changed “…steroid hormones are also...” to “...ovarian steroidogenesis is also…”
Point 7: Line 42: taking into account the course of steroidogenesis the order of these genes should be as follows: “StAR, CYP11A1, 3β-HSD, and CYP19A1”. This order should be continued of kept in the whole text of the paper. Moreover, in the line 42 the authors needs to explain the names of these genes or proteins. For instance, 3β-hydroxysteroid dehydrogenase (3β-HSD), etc.
Response 7: Thank you very much for your advice. We have looked through the whole text and changed the order of key steroidogenesis-related genes. Moreover, We have explained the names of these genes or proteins: steroidogenic acute regulatory protein (StAR), cytochrome P450 family 11 subfamily A member 1 (CYP11A1), 3β-hydroxysteroid dehydrogenase (3β-HSD), and cytochrome P450 family 19 subfamily A member 1 (CYP19A1).
Point 8: Line 45: It should be as follows: Progesterone (PR), androgen (AR) and estrogen (ER) receptors are located…”
Response 8: Thank you very much for your advice. We have changed to “Progesterone (PR), androgen (AR) and estrogen (ER) receptors are located…”
Point 9: Lines 58-59: “It should be as follows: miRNAs act as negative modulators of gene expression.”
Response 9: Thank you very much for your advice. We have changed to “miRNAs act as negative modulators of gene expression”.
Point 10: Line 68: change “mir-31” to “miR-31”
Response 10: Thank you very much for your advice. We have changed “mir-31” to “miR-31”.
Materials and methods
Point 11: Lines 90-98: Here is described the process of isolation of ten different groups of follicles from 6 hen ovaries. Please describe isolation of granulosa layer and theca layer from these follicles since in the Fig. 1 there are presented results of gga-miR-449b-5p expression in the granulosa and theca layers of different groups of ovarian follicles.
Response 11: Thank you very much for your advice. I have added the method of isolation of granulosa layer and theca layer from these follicles in the sample collection section. “The obtained groups of follicles were placed in PBS buffer containing 3% double antibiotics to remove the residual connective tissue and attached blood filaments. The outer membrane layer was peeled off with curved forceps, the follicles were cut in half with scissors and the end was gently shaken with forceps until all the white granulosa layer came out,the remaining layer is the theca layer.”
Point 12: Lines 122-123: Table 1. Change order of the steroidogenic genes as follows: StAR, CYP11A1, 3β-HSD, and CYP19A1.
Response 12: Thank you very much for your advice. We have changed the order of key steroidogenesis-related genes as follows: StAR, CYP11A1, 3β-HSD, and CYP19A1.
Point 13: Lines 153-156: the order of steroid hormones should be as follows: progesterone (P4), testosterone (T), and estradiol (E2) accordingly to course of the steroidogenesis process. Please exchange “androgen” into “testosterone”. Moreover, more data concerning sensitivity of the methods, antibody cross-reactivities, and inter- and intraassay coefficients of variations need to be included.
Response 13: Thank you very much for your advice. We have changed the order of steroid hormones as follows: progesterone (P4), testosterone (T), and estradiol (E2). Moreover, I have also added the following information: “The sensitivity of ELISA Kits of P4, T, E2 were typically less than 10 pmol/L, 1.0 pg/mL and 0.1 pg/mL; tolerance within batch and tolerance between batches of CV < 10% and no cross-reactivity for all three ELISA kits.”
Results
Point 14: In the Fig. 1a. are presented data on expression of gga-miR-449b-5p in different weeks of age, however, in the Materials and methods there is no information that samples were collected from chickens in different age. Therefore, this Fig.1a should be removed.
Response 14: Thank you very much for your advice. We have removed Fig. 1a.
Point 15: Fig. 3a. Change order of the genes in this Fig as follows: StAR, CYP11A1, 3β-HSD, CYP19A1, FSHR, LHR.
Response 15: Thank you very much for your advice. We have changed the order of the genes in Fig. 3a as follows: StAR, CYP11A1, 3β-HSD, CYP19A1, FSHR, LHR.
Point 16: Fig. 3b. Change order of these figures as follows: P4, T, E2.
Response 16: Thank you very much for your advice. We have changed the order of the genes in Fig. 3b as follows: P4, T, E2.
Point 17: Fig. 2, 3, 4 and 5. Provide statistical information concerning meaning of “*” and “**”.
Response 17: Thank you very much for your advice. I have provided statistical information on the meaning of " * " and " * * " in the corresponding places in the original text.
Point 18: Fig. 3. Chart description: gga-miR-449b-5p regulates steroid secretion from GCs of small yellow follicles (6-8 mm). The same remark concerns the other figures.
Response 18: Thank you very much for your advice. I have amended the description of Fig. 3 in response to your comments, and the same comments can be found in Fig. 2 and Fig. 5.
Discussion
Point 19: Lines 297-308: Provide the appropriate order of steroid hormones and steroidogenic genes.
Response 19: Thanks for your advice. I have made the corresponding changes on lines 297-308.
Point 20: Lines 303-304: It should be as follows: “…we determined P4, T and E2 secretion by ELISAs as well as the expression of …by qPCR.”
Response 20: Thank you very much for your advice. We have changed to “…we determined P4, T and E2 secretion by ELISAs as well as the expression of …by qPCR”.
Point 21: Lines 307-308: It should be as follows: “…with the previous results concerning the validation of…”
Response 21: Thank you very much for your advice. We have changed to “…with the previous results concerning the validation of…”
Point 22: Lines 309-321: In this paragraph the authors discuss role of IGFBP3 in ovarian steroidogenesis and the effect of gga-miR-449b-5p on this gene expression. Information concerning role of this gene and protein should be included in the Introduction, especially in the aim of the study.
Response 22: Thank you very much for your advice, we have made the following changes based on your suggestion. “. In another previous study, we predicted that IGF2BP3 may be its targeted regulatory gene[20]. At present, the reports on this gene are mainly focused on cell development, proliferation and migration[43, 44], especially in cancer development and progres-sion[45-47], but there are few reports on the reproduction of laying hens. Therefore, in the present study, we explored the regulatory effect of gga-miR-449b-5p on cell prolif-eration and steroid synthesis and hormone secretion in GCs and explored the rela-tionship between gga-miR-449b-5p and IGF2BP3 in GCs to further explore the role of gga-miR-449b-5p in ovarian function in laying hens.”
Point 23: Abbreviations: It does not comprise all abbreviations.
Response 23: Thank you very much for your advice. We have looked through the whole text and added the following: “StAR: steroidogenic acute regulatory protein; CYP11A1: cytochrome P450 family 11 subfamily A member 1; 3β-HSD: 3β-hydroxysteroid dehydrogenase; CYP19A1: cytochrome P450 family 19 subfamily A member 1; CCND1: cyclin D1; CCND2: cyclin D2; CDK1: cyclin dependent kinase 1; CDK2: cyclin dependent kinase 2; CDK6: cyclin dependent kinase 6; FSHR: follicle-stimulating hormone receptor; LHR: Luteinizing Hormone receptor; IGFBP4: insulin like growth factor binding protein 4; PGRMC1: progesterone receptor membrane component 1; MMP2: matrix metallopeptidase 2; IGF2BP3: insulin like growth factor 2 mRNA binding protein 3; BMP3:bone morphogenetic protein 3; E2F5:E2F transcription factor 5 .”
